# A Framework of Smart-Home Service for Elderly's Biophilic Experience

**Eun Ji Lee [1] and Sung Jun Park [2],***

1   Department of Architecture, Keimyung University, Daegu 42601, Korea; yej@stu.kmu.ac.kr
2   Department of Architectural Engineering, Keimyung University, Daegu 42601, Korea
*   Correspondence: sjpark@kmu.ac.kr; Tel.: +82-53-580-5765

**Abstract:** Smart-home technology and related services can reinforce a person's experiential nature, promoting sustainable living among the elderly. It is crucial in the housing industry that support "Aging in Place", contributing to the contact, control, and simulation of nature at home as well as the creation of a high-quality living space instead of mechanical achievement. Further, biophilic experience, the strengthening of inherent human propensity to nature for optimal health and well-being, supports the elderly's physical, mental, and sociological health. However, despite the continuing emphasis on the benefits of residential nature experiences for the elderly, the application of smart-home technology and services is insufficient. This study presents a theoretical basis for combining biophilia and smart-home technology, providing a framework for smart-home services to ensure elderly residents can have biophilic experiences. In this study, smart-home components and related studies that can support the biophilic experience and the corresponding technology are analyzed. The results suggest the type and content of smart-home service for ensuring a biophilic experience, while also indicating the configuration of supportive input and output devices according to the service framework. Moreover, we recommend the interaction characteristics of smart-home devices from the perspective of residents, space, efficient service provision, and physical application. This paper broadens our understanding of the sustainable, residential-environment nature experience and informs the expansion of the aged-friendly smart-home industry, contributing to smart-home services trends and development.

**Keywords:** elderly; biophilia; biophilic experience; smart home; smart-home service; service framework

## 1. Introduction

The most important factor in the residential space for the elderly is their experience, and the main objective is to induce and support experiences related to health and well-being. The priority of the elderly experience in a residence should take into account the fundamental needs of the elderly and the resulting benefits. In particular, it is very important for the elderly to maintain constant contact with nature in their physical environment and to stay healthy. "Biophilia" is a concept that explains the relationship between humans and nature and is defined as "the inherent need among humans to interact and mingle with nature in order to achieve and maintain their optimal health and well-being" [1]. Biophilia has a positive impact on the elderly in terms of productivity and emotional well-being, stress reduction, learning, and recovery [2,3]. Moreover, related studies have mentioned that biophilia provides practical help to the elderly, which can be measured or proven [4–7]. Accordingly, biophilic design is being applied to medical and nursing facilities for the elderly [8], and the active attitude of the elderly toward successful aging and the pursuit of "Aging in Place" (AIP) emphasizes the need to connect with nature in residential spaces.

AIP has been a popular topic in social welfare academia in recent years, and it is deemed meaningful because it forms a community care service network in residential spaces that considers economic and efficiency aspects and allows aging with familiar people in familiar areas [9]. It should be noted that services supporting AIP are developing with technological advancements. As the "baby boomers", people who were born in the period of markedly increased population, are integrated into the aging population, the conventional concept of the "elderly lifestyle" is changing, showing a different pattern from that of the previous generation [10]. Currently, smart technology occupies a large part of the elderly's daily life, and the concept of "smart aging", which considers the acceptance and understanding of technology by the elderly, has a great influence on the intelligent housing industry [11]. Smart homes for the elderly focus on providing efficient healthcare and convenience, such as real-time monitoring systems and remote medical treatment, fall detection and response, security, and safety management. Even though biophilia has long explained the importance of nature on health and welfare [1,2,12–15], few studies have investigated smart-home services that support and connect the elderly to nature. Contemporary smart healthcare services are useful for treating diseases and responding to emergencies; however, planning techniques that expose the patients and the elderly to a healthy natural environment could be more important, as they can provide the opportunity for the elderly to maintain a good health. This is a problem that requires adequate and thorough research. The natural environment closely affects human health, and its deprivation causes disorders such as fatigue, depression, high blood pressure, diabetes, and cancer [16–18]. Therefore, from a preventive perspective, it is necessary to find a methodology that can induce biophilic experiences to help the elderly living in urban environments, where the natural environment is scarce, to maintain a healthy life.

The purpose of this study is to suggest a theoretical basis for combining biophilia and smart-home technologies and to provide a framework by designing smart-home service content for the elderly in residential spaces. The concept of biophilia is not contrary to industrial and urban development, and it can act as a catalyst for key elements of building-related technologies. Smart lighting, heating, ventilation, and air conditioning (HVAC) control systems in the field of architectural planning are elements that can maximize contact with nature, and the use of Internet of Things (IoT), Virtual Reality (VR), and Augmented Reality (AR) technology enables an increase in the effectiveness of natural production in a limited physical environment. The health benefits associated with experiencing nature arise from the opportunity to perceive it rather than from direct contact [19], and this includes natural analogs such as shapes, colors, and sounds that can remind us of nature through images [1,8,19]. In addition, recent studies have demonstrated the healing effect of the natural environment based on virtual reality [20–22]. In other words, the use of IoT can help resolve the problems of physical and spatial limitations and the physical environment in residential spaces, enabling a satisfying experience with nature. Figure 1 shows the scheme used in this research.

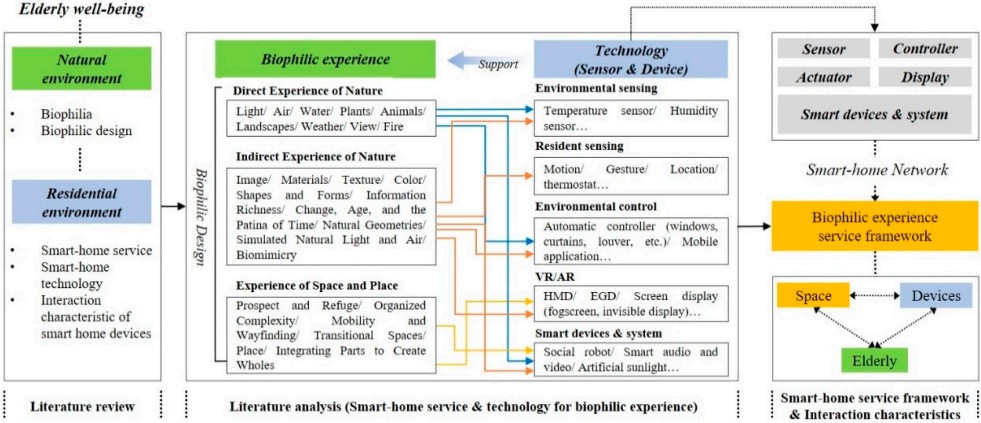

**Figure 1.** Research scheme.

First, to achieve the aim of this study, current research is reviewed in relation to the natural environment and in terms of the effects of biophilia on the health of the elderly, i.e., the concept of biophilic experience. In addition, the existing smart-home service fields and components are analyzed, focusing on studies related to the residential environment and smart-home service for the elderly. Second, through literature analysis, the link between the key elements of the biophilic experience [23] and the smart-home services and technologies related to the health of the elderly are systematically analyzed according to three categories. The literature analysis of this study was performed through major academic databases, including Scopus, IEEE Xplore, SpringerLink, ScienceDirect, and PubMed, and is described in detail in Section 3. Third, we propose a service framework by constructing the contents of the biophilic experience-based service of the elderly with regard to smart-home technologies and devices. Finally, the input and output devices that can support biophilic experience services according to the service framework are listed. Additionally, the interaction characteristics of smart-home devices are proposed from the perspective of residents and in terms of the available space, while also considering efficient service provision and physical application in the house.

Previous studies have focused on combining architectural elements concerning physical spatial planning, based on the concept of biophilia. This study is novel because it seeks to link smart-home services and biophilic experience in the realization of AIP and the change in perception of the elderly, targeting residential environments that lack exposure to nature. This study highlights the need for smart-home technology to support experiences with nature for a sustainable living environment for the elderly and discusses its potential. Human-centered smart-home services based on biophilic experience play a fundamental role in inducing active and independent living of the elderly. The results of this study inform the expansion of the field of the aged-friendly smart-home industry and contribute to the theoretical basis for encouraging the biophilic experience and the development of smart-home services and technology.

## 2. Theoretical Insights and Background

### 2.1. Aging and the Natural Environment

According to the World Health Organization (WHO), health is defined as having complete physical, mental, and social well-being as well as being free from disease and functional impairment [24]. Advances in science and medical technology have enabled the treatment of various diseases and have extended life expectancy; however, this does not guarantee a "healthier life." The essence of the aging problem in a global aging population is not that older people live longer, but that they do not live a healthy lifestyle. In old age, they experience not only certain diseases, but also uncomfortable conditions such as chronic fatigue, pain, depression, and sleep disorders [25]. Financial status, income level, and social relations tend to decline after retirement, while emotional complications increase [26]. In addition, a decrease in sensory functions such as visual acuity, hearing, and touch can also affect agility, balance, muscle strength, and endurance, as these are closely related to cognitive and mental health as well as physical health [27,28]. Specifically, the decline in cognitive and mental health of the elderly affects memory, language function, and information processing speed, and the elderly may experience avoidance, anxiety, and lethargy, accompanied by mental stress and thus may be subject to a relatively high risk of chronic depression [29]. Therefore, it is important to alleviate or delay sensory damage with aging and to improve the independence and quality of life (QoL) of the elderly; thus, it is necessary to establish a residential environment from a healing perspective.

"Healing" can be defined as moving toward a healthy state through psychological, environmental, cultural, and social support based on experience, and it is distinguished from "curing", which recovers diseases through medical means [30]. From an environmental perspective, healing involves not only sight, but also various human senses such as touch, hearing, and smell, simultaneously, and space is recognized through multi-sensory reactions and experiences of mental and psychological changes such as emotional stability [30]. In the case of the elderly, a richer multi-sensory environment should

be provided by weakening the sensory function of experiencing space, and many previous studies have emphasized the connection between nature and a healing environment for the elderly [31–33]. In the field of environmental psychology and neuro-architecture, the benefits of nature and how it affects individuals' health and well-being has been highlighted for centuries [34], yet now it requires more attention amid the increase of the aging population and population concentration in urban areas. Exposure to nature and natural analogs leads to positive rather than negative emotions and thoughts, and these changes have been demonstrated through responses such as blood pressure, heart rate, and muscle and brain activity [2]. Marcus [35] categorized natural healing effects into three categories. First is relief from negative physical and biological symptoms or at least relief from the perception of the symptoms. Continuous reminding of one's illness and pain can lead to a negative psychological state, and since nature stimulates the five senses to induce recovery through involuntary attention, aging-related symptoms can be alleviated [36]. Second is the stress reduction and recovery of cognitive function. The human brain responds to sensory patterns and elements found in the natural environment, and concentration and thinking skills are improved when humans interact with nature, and factors related to memory loss are suppressed [8]. Third is the improvement of awareness of well-being and satisfaction. As elderly people experience multidimensional aspects of aging, they have relatively low health-related consciousness, such as satisfaction with their subjective health status and quality of life, and they are more vulnerable to physical environments that negatively affect their health [37]. In other words, the conscious negative psychology arising from the individual's health and the surrounding environment can substantially contribute to the quality of life of the elderly through the connection with natural factors. This study emphasizes that the residential environment of the elderly should be planned with the intention of recovery and healing through the connection with nature as a physical measure to manage and prevent aging problems.

Table 1 shows the contents and results of previous studies that investigated the benefits of nature for the elderly. The effects of nature related to the health of the elderly vary physically, mentally, and socially and appear as physical function enhancement and recovery, psychological recovery, emotional satisfaction, and an increase in the subjective health and welfare index. In particular, horticultural programs such as plant cultivation and harvesting can serve as an economic motivator for productive activities in old age. The natural environment and elements covered in previous studies include not only direct-contact activities, but also indirect nature-inducing materials such as images and videos related to nature. The simulated natural environment showed a recovery effect similar to that of the real natural environment. The characteristics of natural experiences that contribute to the health and well-being of the elderly are direct and indirect views of specific landscape elements, such as green space and water, and various vegetation. This creates a multi-sensory environment that represents changes in nature, such as sunlight and shade, as well as the sounds of nature. It includes providing activities and environments that involve the communication with living organisms (animals and plants) in nature, while maximizing repeated access to experience these characteristics. This expands our understanding of how to experience nature and suggests the potential value of actively utilizing the benefits of nature in physical space.

**Table 1.** Health benefits of nature associated with the elderly.

| Resource | Participants | Measurement | Outcomes | Health Benefits |
|---|---|---|---|---|
| [38] | Over 50 years old. $n = 103$ | : Visit the park located in the city center for more than two hours a week for one year : Survey of natural factors and environment of parks and awareness of health and nature experiences (Likert scale 7) | : The higher the awareness of natural experience, the lower the level of anxiety about health, the higher the level of physical function, and the higher the preference for natural elements | Physical function enhancement/ psychological recovery/emotional gratification |
| [39] | 62–93 years old $n = 50$ | : Comparison of the survey results on the natural image of the non-elderly and the elderly/the elderly are divided into three groups according to their living environment, such as cities, rural areas, and facilities : Conducts a familiarity, preference, and resilience assessment of images in environmental categories: streets, cities, and natural environments, etc. (Likert scale 11) | : For all three groups of senior citizens, hill and lake images are considered to be more resilient, preferable, and the most familiar compared to residential, urban, and industrial areas : The correlation between preference and resilience is statistically significant in the elderly group | Psychological recovery/emotional gratification |
| [40] | Over 65 years old $n = 61$ | : The experimental group is exposed to sunlight for about 2 h for 5 days : Measure the demographic status, physical condition (blood pressure and pulse, chronic disease), and PSQI of the elderly | : The sleep quality score of the experimental and control groups improved from 10.45 +/− 1.98 to 6.081 +/− 2.45 after exposure to sunlight ($p < 0.001$) : Strong positive relationship between sun exposure time and sleep time, regular sleep activity, and sleep state | Positive biorhythm and sleep quality |
| [41] | Under 79 $n = 53$ | : Survey and in-depth interviews on subjective physiological and emotional responses to soundscape | : The closer the sound of nature, the more positive and preferred it is : Prefer the sound of birds and leaves, and recognize the sound of waterfalls, animals, etc., as pleasant sounds. | Emotional gratification |
| [42] | The elderly in Tokyo $n = 3144$ | : Analysis of the correlation between the life expectancy of the elderly and the natural environment around their residence (for 5 years) : Survey on the conditions of living environment excluding social and economic effects of the elderly (self-response formula) | : The positive living conditions for the longevity of the elderly are the creation of parks and street trees, the absence of noise from cars and factories and the area and time of high sunlight. | Higher life expectancy |
| [43] | Adults and the elderly $n = 30$ | : Participate in outdoor garden and indoor reading activities for 30 min after performing stress-inducing tasks (Stroop) : Repeated measurements of saliva cortisol readings and self-reporting assessments | : Both garden walks and reading activities resulted in a decrease in cortisol, but the garden walk group showed a much lower level of decline. : In particular, stress levels have fully recovered after gardening, but there are more cases of recital after reading activities. | Reduced stress/psychological recovery |
| [44] | Alzheimer's patient $n = $ not presented | : Observation and investigation of behavioral disorder characteristics of Alzheimer's patients using closed gardens | : Patients visiting the garden show reduced behavioral disorders and reduced anger control behavior | Emotional/psychological recovery |
| [45] | Alzheimer's patient $n = 28$ | : Monitoring medication type, dosage, falls, etc., according to the frequency of use of the patient's garden | : The number of falls and the severity of falls of older people with more frequent garden use decreased by approximately 30% : Significant decrease in anti-psychotic medication | Physical function enhancement/ psychological recovery |

**Table 1.** *Cont.*

| Resource | Participants | Measurement | Outcomes | Health Benefits |
|---|---|---|---|---|
| [46] | Average age 73 $n = 193$ | : Preference analysis of green space characteristics in urban environments related to the subjective stress index<br>: Accessibility and noise level of green park and preference for the presence of shade, ponds, aquatic organisms, etc. (visual discrete choice experiment) | : Prefer green space that is easily accessible and quiet and cool<br>: The more shade, trees, and green spaces with ponds there are, the higher the preference. | Emotional gratification |
| [47] | A resident around a green space. $n = 1347$ | : Conduct random selection and post-survey<br>: Frequency of activities in green spaces by age, preference for characteristics of natural environment, WHO welfare index<br>: Older = 55–65 (21.4%), 66+ (18.3%) | : Prefer spaces that nature-like/rich in species/lush/beautiful/varied<br>: The higher the frequency of activities in green spaces, the higher the preference for natural environment characteristics and the WHO welfare index | Emotional gratification/ subjective health and welfare index rise |
| [27] | The elderly $n = 28$ | : Survey on physical and mental effects of plant-based gardening treatment programs.<br>: Participates in horticulture once a week for two months<br>: Cortisol measurements and senior fitness tests for one week before and after the program ends | : The cortisol levels in the horticultural therapy group significantly decreased after horticulture activities compared to the levels before<br>: SFT's 6 sub-check item scores have been greatly improved | Physical function enhancement/reduced stress |

Note: PSQI = Pittsburgh Sleep Quality Index, SFT = Senior Fitness Test.

## 2.2. Biophilic Experience and Biophilic Design

### 2.2.1. Biophilia and Biophilic Experience

The relationship between nature and humans can explain humans' general preference for nature from an evolutionary perspective based on early geographic research. An individual's feelings and perceptions related to environmental preferences are based on the characteristics of places that are likely to overlook the landscape as well as places that you can discover resources or hide from dangers or threats [48]. Biologist Edward O. Wilson further advocated the theory of "biophilia" from an evolutionary point of view and has expanded this concept. Biophilia, which refers to human intrinsic affinity for life and natural systems, discusses the relationship between nature and humans based on the emotional alliance of humans inherent in natural life [1]. The goal of biophilia is to restore a healthy relationship between humans and the environment, and related studies have explored biophilia as a method to induce positive natural experiences, while offsetting negative environmental factors and creating a relaxed mental state [23,49].

Human indoor occupancy time is increasing, and for the elderly, social activity decreases due to aging and, in particular, retirement. Such daily life isolated from the outdoor environment cuts off positive experiences with nature and damages the relationship between humans evolutionary and instinctive characteristics with nature [17]. However, experiences with nature based on biophilia can provide a wide range of physical, mental, and behavioral benefits. Physically, these include reduced blood pressure, increased comfort and satisfaction, decreased disease symptoms, and improved physical health [2,35,45]. Mental benefits include satisfaction and motivation, reduction of stress and anxiety, and an improvement in problem-solving skills and creativity [8,36,50]. Furthermore, positive behavioral changes increase attention and social interaction, while decreasing hostility [23,44]. Therefore, informing a planning technique that enhances biophilia factors in indoor spaces is a research issue that should be addressed with great importance, as it can be a solution to various health problems.

Kellert and Wilson [1] argue that biophilia can be achieved not just by an instinct, but by complex learning rules and repetitive experiences that cultivate and factor in human biological tendencies. In other words, the biophilic experience cannot be replaced by physical concepts or objects such as

architectural styles or advanced devices and technologies that have appeared in the process of the development of modern society; however, it can be appropriately expressed in a new environment that is conscious and artificial and continues in future generations [51]. This study focuses on the method and possibility of linking the positive natural experiences of the elderly with smart-home technology based on the biophilia learning rule, that is, the biophilic experience claimed by Wilson.

### 2.2.2. Biophilic Design

Biophilic design is a design strategy that aims to make modern living spaces suitable for promoting human health and well-being [8]; an applied science that aims to plan a physical environment that reflects the concept of biophilia. Biophilic design stemmed from sociology and ecology in the study by Kellert [8] and has been addressed in various fields such as psychology, medicine, and architecture since then. Biophilic design differs in characteristics and classification criteria depending on the relevant researcher and has been embodied in dimensions, elements, and attributes of biophilic design [8], patterns of biophilic design [49], experiences and attributes of biophilic design [23], and the biophilic quality index (BQI) [52]. Practical examples of biophilic design measurement criteria based on this are the Living Building Challenge (LBC) and the WELL Building Standard (WELL) in the U.S. The LBC is a certification system developed by the International Living Future Institute (ILFI), referring to six elements and 72 attributes of the biophilic design proposed by Kellert [53], and it requires a high level of environmental performance. The WELL Certification, developed by the International WELL Building Institute (IWBI), is the first certification system focused on human health and comfort. The criteria for measuring biophilia in this certification are addressed by the "mind concept", which emphasizes the improvement of users' mental health through construction programs and designs [54]. Although biophilia is only one essential component, and there are additional points in obtaining certification, it is an important factor that has a unique impact on the project.

Kellert [23] emphasized the importance of the biophilic experience for practicing biophilic design and organized it according to nine biophilic design principles and three categories: Direct Experience of Nature, Indirect Experience of Nature, and Experience of Space and Place, all of which propose to help integrate the relationship between nature and human into the architectural environment. The details are shown in Table 2.

**Table 2.** Experiences and attributes of biophilic design by [23].

| Direct Experience of Nature | Indirect Experience of Nature | Experience of Space and Place |
|---|---|---|
| Light<br>Air<br>Water<br>Plants<br>Animals<br>Weather<br>Landscapes<br>View<br>Fire | Images<br>Natural materials<br>Simulating natural light and air<br>Naturalistic shapes and forms<br>Evoking nature<br>Information richness<br>Age, change, and the patina of time<br>Natural geometries<br>Biomimicry | Prospect and refuge<br>Organized complexity<br>Integration of parts to wholes<br>Transitional spaces<br>Mobility and wayfinding<br>Cultural and ecological attachment to place |

The study by Kellert [23] underlines a set of options for enabling the biophilic experience in an effective way, replacing a limited checklist of options; these options can be appropriately and specifically applied to the presented experiences and attributes. In other words, the artificial environment pursued by biophilic design is not limited to indoor spaces, and new perspectives and meaningful attempts for appropriate and specific application methods are needed. Until now, previous studies related to biophilia and biophilic design have played a significant role in advancing our understanding of how the physical environment affects health and well-being; however, most of the studies have not considered the variety of smart methodologies and technical aspects of experiencing nature. Specifically, there is a

tendency among them to overlook everyday places such as residential homes in modern society and urban environments [34]. This paper seeks a fresh perspective and an appropriate plan to enable the biophilic experience in smart-home technology and services, based on previous studies.

*2.3. Residential Environment and Smart-Home Service for the Elderly*

In recent years, the concept of "successful aging" has emerged, a concept that seeks methods for supporting multi-dimensional needs and possibilities for the elderly in social, psychological, and biological aspects to address such needs. In the past, spatial design, which originally focused on care facilities and medical centers for the elderly, has been expanded into housing and communities to support their daily lives [55]. In particular, among the methodologies for realizing successful aging, AIP, which involves getting older with familiar people in a familiar area, has important significance in the expectation that caring for local communities is cost-effective in terms of reducing financial burden [9]. Global aging and a change in the life consciousness of the elderly have had a significant influence across industries such as society, science, culture, and environment, and there is a consensus that focuses on having an interest in the residential environment of the elderly, which to date has been neglected in the existing welfare policy that supports AIP [56]. That is, the residential environment for the elderly is the key to realizing AIP; a positive, healthy residential space is crucial for their well-being and is needed for them to have a productive life. Lawton [57] argues that from a person–environment fit perspective, the life of the elderly varies according to the fit between the individual's situation and the environment, meaning that even within the same type of environment, people who are more vulnerable in terms of health, function, and resources are more affected by the environment than others. Therefore, elderly people must live in a place with environmental conditions that can promote their health and ability to function; thus, natural factors that can support the elderly's multi-dimensional health [38,39,43–47] should be applied to the elderly's residential environment.

These days, support for residential-based services has been developing so that elderly people can lead independent daily lives in the local community and live a prosperous life. As various innovative technologies are combined with urban infrastructure, the interest and demand for smart homes for the elderly are rapidly increasing. In addition, as housing is recognized as an important resource for elderly care, along with the AIP concept, providing services not only for health management but also for safety, convenience, and daily life support has become increasingly important. Smart homes refer to houses equipped with home networking technology and systems to improve the quality of life of the residents [58], and studies on customized smart-home service environments considering various social classes have been actively conducted recently. Previous studies related to smart homes for the elderly underlined self-sustaining and sustainable service environments, such as home automation and the convergence of home-appliance-based comprehensive living convenience, health care, resident management, and self-supporting residential systems [59–61]. In a situation where most smart-home environments consist of device–device and device–occupant interaction services, which centers on home appliances and individual devices, the interaction method of smart devices is related to the input/output method of information, thereby affecting the usability of the elderly [62]. This is true among various groups and is also closely related to spatial characteristics such as the shape and size of the devices, for example, how the devices are installed, etc. [63]. Therefore, this study considers the technologies that make up the smart-home service as well as the interaction characteristics of devices from the viewpoint of the residents, while also considering the spatial perspective, as shown in Table 3.

With respect to the interaction with the occupants, the contact type refers to control panels and switches that users use through direct interaction, while non-contact refers to sensors and receivers that perform their functions in the living environment of occupants without direct contact [64]. From a spatial point of view, non-contact is divided into mounted (M) and embedded, both of which require additional installation and attachment in the space; of these, the embedded type includes both space embedded (SE), that is, built into the spatial structure, and equipment embedded (EE) for which service-specific functions are included in the smart devices or intelligent devices replace existing ones [65].

Table 3. Interaction characteristics of the smart-home devices and equipment.

| Type of Interaction | | Characteristic |
|---|---|---|
| Resident | Contact | Requires direct interaction and physical contact with residents (touch screen, smart floor, etc.) |
| | Non-contact | To perform functions within the living environment without physical contact or physical manipulation of residents (automatic controller, luminance sensor, etc.) |
| Spatial | Mounted type (M) | Connect additional devices to control service functions or attach them to existing environments (attachment type sensor, remote controller, etc.) |
| | Space Embedded (SE) | Built into spatial structure, invisible (transparent switchable glass, smart windows, wall display, etc.) |
| | Equipment Embedded (EE) | Service-specific features are built into smart devices or replaced by intelligent equipment (mobile application, intelligent objects, social robot, etc.) |

Smart-home services consist of sensors, controllers, actuators, displays, and other smart devices and systems as a network, enabling the localization and remote control of domestic environments as well as automation [62,66]. The home network gathers and stores data transmitted from the input device through the home gateway or platform and converts it into a single protocol to deliver it to the output device [67,68]. Smart-home services for the elderly aim to strengthen physical, mental, and social health management of the elderly and can be categorized into health monitoring (HM), environment monitoring (EM), risk management (RM), awareness improvement (AI), and community management (CM) [60,69]; details are shown in Figure 2.

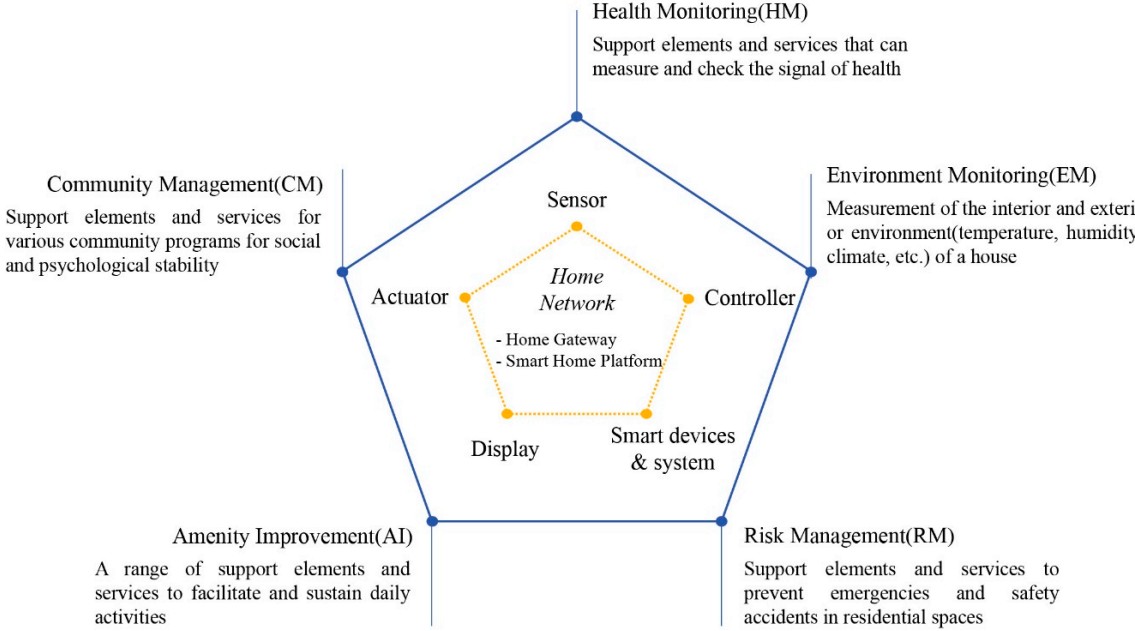

**Figure 2.** Smart-home service components and contents.

Smart-home technology is being used to support the various needs of the elderly, and the fundamental goal of all technologies and services is to improve the QoL of humans [60]. From this perspective, although smart-home services maximize the convenience and efficiency of housing for the elderly and families, it is often seen as a "mechanical utopia" rather than a high-quality living space. Up to now, smart homes for the elderly have focused on checking their health status in real time,

responding to crisis situations as quickly as possible, and communicating more easily with medical professionals or family members [70]. Importantly, despite the continuing emphasis on experiences with nature in the residential environment of the elderly, discussions about applying it to smart technology or turning it into a service are insufficient. This stems from the perception that nature and smart-home technology are contradictory concepts; another reason is that smart-home-related research usually focuses on the modernization of existing smart-home service systems. However, the development of smart-home technology and its devices to date can be effectively used in contacting, controlling, and directing nature within the home environment, and the concept of biophilia based on the relationship between humans and nature could lead to a new trend in the smart-home industry; thus, biophilia must be considered. Therefore, the smart-home service for the elderly should focus on supporting a more multi-sensory nature experience and immersion in the experience by easing the limitations of the physical, regional, and geographical conditions, and should provide a high-quality residential environment by connecting nature and technology.

## 3. Services and Devices for a Biophilic Experience

This study has considered previous studies regarding smart-home services and technologies for the elderly to integrate technologies and services that support the elderly's experiences of biophilia in their homes. To systematically review smart-home services and technologies for the elderly, and support their experiences with biophilia, a variety of keywords were used, and in the case of the selection criteria for services and technologies, we refer to the analysis contents of Tables 1 and 2 specified in Section 2. Table 4 shows the details.

**Table 4.** Search strategy for literature review.

| Analysis Framework | | Contents |
|---|---|---|
| Keyword | Elderly | "Aged", "Old people", "Aging population", "Senior", "Older adult" |
| | Services and technology | "Smart-home service", "Smart-home technology", "Healthy living", "Gerontechnology", "Biophilia technology", "Biophilic design industrial", "Indoor garden technology", "Ubiquitous technology", "Robotics", "IoT", "Automated home", "VR (Virtual Reality)", "AR (Augmented Reality)", "Immersive technology" |
| Criteria | | - Does it contribute to the health and well-being of the elderly?<br>- Does it support direct and indirect views in relation to specific natural environmental factors?<br>- Does it support multi-sensory experiences of various changes in nature such as light, wind, sound, etc.?<br>- Does it support direct and indirect communions with natural creatures?<br>- Does it support experiences with natural ecosystems and systems?<br>- Does it support accessibility for repetitive experiences of those features? |

The initial keyword search resulted in 45 duplicates out of 303 papers in six databases. After that, 124 documents were screened, excluding 98 documents that were judged to be inconsistent with the research topic and analysis criteria. After screening the final full-text review, 93 papers that did not meet the criteria were excluded, such as accessibility and application of housing and relation to nature or ecosystems. Finally, 41 papers were selected according to the eligibility criteria. Figure 3 shows the details of this procedure.

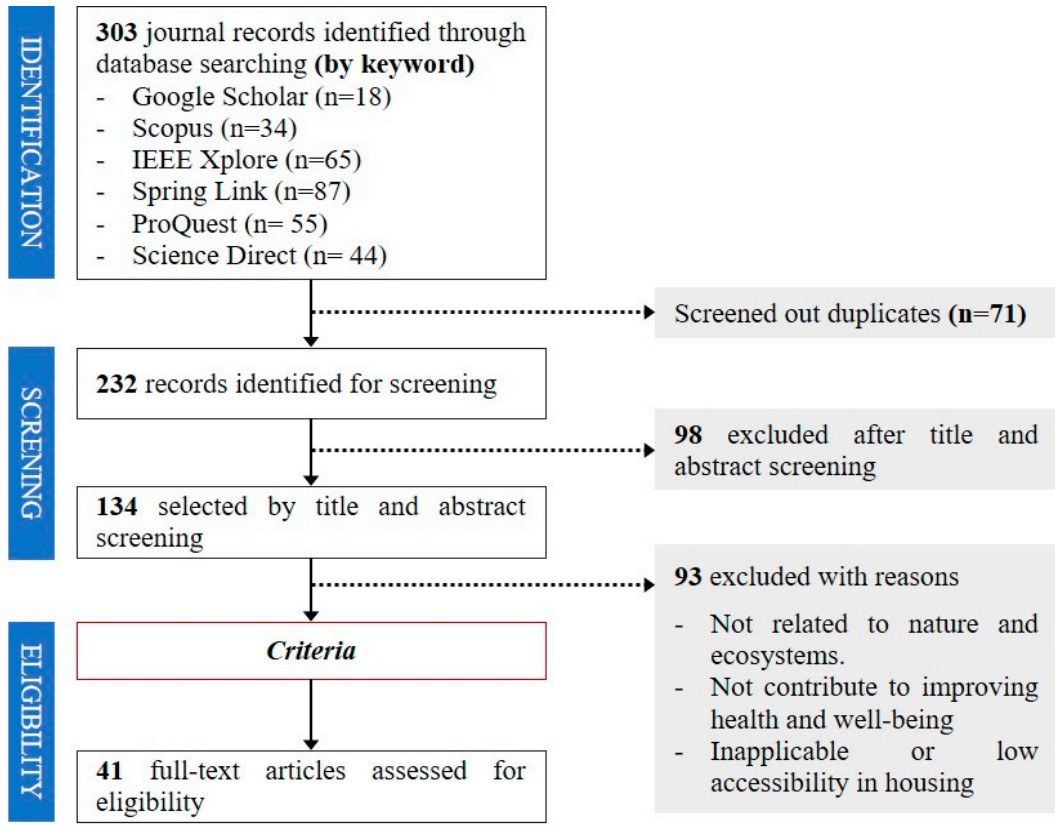

**Figure 3.** Literature review flow diagram.

Based on the literature reviewed, this study derived the types of smart-home services that involve experiences with biophilia according to Kellert's three categories of experiences [23], proposing sensors and devices according to the configuration of smart-home services.

### 3.1. Direct Experience of Nature

The direct experience of nature is fundamental to having experiences with biophilia, which represents an actual contact with the features and characteristics of the natural environment [23]. Lighting and air quality in homes, which are key factors that improve the quality of life, are already being provided by an environmental control system in the smart-home industry. However, existing environmental control systems that focus only on energy efficiency and convenience from a bio-friendly perspective need to provide us with experiences of natural light and air in a more diverse and creative way. Along with automatic opening and closing devices, supporting devices, such as louvers and reflectors that can track the sun's angle and its light direction (Heliostat Mirror) [23] and diffuse or transform it, are required to provide various light environments. In addition, a single home network should also be controlled so that the devices can achieve an optimal lighting environment. In particular, responding to weather conditions generating an awareness of changes in the seasons and weather can also be expressed as the potential needs of humans [71]; allowing users to see what is outside the building in real time gives them a sense of protection and understanding of nature. This includes collecting solar and rainwater and listening to the sound of wind or rain [23]. Communing with living things is another factor that induces us to have direct experiences with nature. Considering the results of previous studies, which found that spending time with companion animals and performing horticultural activities is beneficial to the mental health of the elderly and promotes their social activities [27,72,73], it is considered desirable to provide companion animals and an environment in which plants can be cultivated in the elderly's homes with smart devices and

systems for easy management and care. As a relevant case, Japan recently developed an animal-shaped artificial intelligence (AI) robot to help alleviate the feeling of isolation and depression in the elderly, allowing them to interact with animals and receive comfort from them even without the biological care required of actual pets [72]. Indoor fish tanks are an effective way to observe living things and access water in the house, and using a smart aquarium based on IoT makes it possible for users to easily utilize the benefits of living things and water, such as water quality and food management, proper lighting intensity control, etc. [74]. A smart plant grower, a product developed for use in homes or small farms with the development of smart agriculture, can act as an artificial supply of light, moisture, soil, and temperature necessary for plant growth, helping to maintain the best plant conditions while spending little time and effort to manage it; this has recently resurfaced due to the increasing interest in indoor landscaping and food safety [75]. In particular, environmental control technologies regarding plants are mainly used to improve air quality, and with the recent development of the greenwall irrigation automation system (GIAS), which consists of low-cost microcontrollers, microprocessors, and applications [76] it has become possible to provide natural and comfortable air quality at a low cost. Table 5 summarizes the services and technologies that support direct experiences with nature in the house.

**Table 5.** Smart-home services for direct experience of nature.

| Service | Devices and Sensors | | Literature Source |
|---|---|---|---|
| All services | Environmental controller and actuator | Automatic controller and actuator (windows/louver/curtain/HVAC system/other smart devices and appliances, etc.) | [77–79] |
| Maximization of light | Environmental sensor | Luminance sensor | [77,80] |
| | Environmental controller and actuator | Heliostat/mirror | [23] |
| Pleasant air and thermal | Environmental sensor | Temperature/humidity sensor | [77,80,81] |
| | | Air volume/direction sensor | [79,81] |
| | | $CO_2$ sensor | [77–79] |
| | | TVOC sensor | [77] |
| | | Dust sensor | [77] |
| | Smart devices and system | Greenwall irrigation automation system (GIAS) [1] | [76] |
| Animals and plants | Smart devices and system | PC/mobile devices | [76,82] |
| | | AI social robot (Aibo/Paro/NeCoRo) | [55,72,73] |
| | | Smart aquarium | [74] |
| | | Smart plants growers | [75] |
| View and weather | Environmental sensor | Video/mike outdoor sensor | [80] |
| | Smart devices and system | Rainwater harvesting system (RHS) [2] | [79,81] |
| | | Three-dimensional (3D) surround speaker | [55] |
| | | Smart glass | [55,83] |

[1,2] Most of the paper sources for the elements of smart-home devices were literature review papers. The terms of devices were attuned to maintain consistent terminology, and these were newly established in this study.

## 3.2. Indirect Experience of Nature

The indirect experience of nature refers to the experience based on an image similar to nature, natural analogs, or its characteristics and processes [23]. In modern city centers and buildings, where it is generally difficult to have direct experiences with nature, the development of technology plays an

important role in expressing the characteristics of nature as if they were real. The indirect experience of nature is a part of certain patterns and processes that appear in the natural world, which include natural textures, colors and shapes, simulations of light and air, and the passage of time [23]. In relation to the biophilic experience, stimulation of the five senses is considered substantially important, and in particular, the health benefits associated with experiencing nature arise from the opportunity to perceive and observe them rather than performing direct activities [36].

Immersion technologies, namely VR and AR, are particularly effective in maximizing the biophilic experience and having experiences with nature through simulated VR and AR environments also enhances physical and mental health conditions similar to that of experiencing real nature [84,85]. The VR and AR for visual stimuli can be implemented through a head-mounted display (HMD), an eye glasses-type display (EGD), a hologram projector device, or a screen-shaped display and are separated into a FogScreen, a wall display, an invisible display, etc., according to the projection method [83]. Of these, the FogScreen is chemical-free and can safely come into contact with clothing and household apparatus, as it is an immaterial projection screen composed of fog and airflow [86]. It comes in two types—either it is a stand-bar type or it is ceiling-mounted, so that it is inexpensive and can form a self-sustaining system through interlocking with a dehumidification system. An invisible display is transparent through which light and users' eyes are transmitted, therefore it can realize realistic AR by reflecting their visual direction to match the synchronization between the physical object and the media [83]. Angelini et al. [87] developed a prototype of a multisensory interactive window for the purpose of supporting the elderly's view of the natural environment and family connection; it is an intuitive interface that allows them to interact with their desired objects at any time and place. In addition, the use of a surround-sound speaker with a smart display allows synesthesia and hearing to be stimulated simultaneously, and the sounds of birds, leaves, wind, streams, waterfalls, etc., can promote the recovery of the elderly; however, the extent of the effect may vary depending on the individual's preferences and circumstances, so that it is necessary to provide sounds according to the information and condition of occupants [88]. In particular, when simulating virtual objects in existing residential spaces, immersion and satisfaction are obtained through smooth interaction [85]. Thus, there is a need to consider a multimodal interface, such as an occupant's voice, gesture, touch, etc., as well as scanning 3D spaces. The LiDAR sensor, which can detect user location and motion, including the scanning of 3D spaces, can precisely track the indoor environment and daily life through laser beams without the need for video equipment, meaning that it has advantages in terms of protecting privacy and maintaining security in the house [89].

Another essential element in the indirect experience of nature is the change in nature over time. Users can experience it through light, darkness, and air flow appropriately simulated according to season and time, which is supported by smart lighting, an HVAC system, etc. In recent years, the artificial lighting (similar to sunlight) and an artificial skylight provided by a virtual sky, etc., have been developed to be applied to houses and facilities for the elderly [90], making it possible to create a natural light environment indoors through projector lamps that reproduce light reflected from water, sunlight through the branches, shadows, etc. [55]. Table 6 shows a smart-home service and technology for the indirect experience of nature.

**Table 6.** Smart-home services for indirect experience of nature.

| Smart-Home Services | Devices and Sensors | | Literature Source |
|---|---|---|---|
| All services | Resident sensor | LiDAR (Light detection and range) sensor (location, movement, activity, behavior) | [78,89] |
| | | Multimodal sensor (voice, touch, gesture, etc.) | [62,82] |
| | | Physiological sensor (sleep, pulse, etc.) | [60,78,82] |
| | Smart devices and system | PC/mobile devices | [60,62] |

**Table 6.** *Cont.*

| Smart-Home Services | Devices and Sensors | | Literature Source |
|---|---|---|---|
| Virtual and augmented nature | VR/AR Display | Wearable display (Head mounted display (HMD), Eye glass display (EGD), etc.) | [73,85] |
| | | Hologram projector | [55] |
| | | Screen display (FogScreen, wall display, invisible display) | [83,86] |
| | Smart devices and system | 3D surround speaker | [55] |
| | | Smart window (multisensory interactive window) | [55,87] |
| Simulating natural light and airflow | Environmental sensor | Luminance sensor | [77,80] |
| | Environmental controller and actuator | HVAC (Heating, ventilation, and air conditioning) system | [60,81] |
| | Smart devices and system | Smart guide lighting | [77,80] |
| | | Artificial sunlight | [23,90] |
| | | Projector lamp | [55] |

### 3.3. Experience of Space and Place

The experience of space and place discusses ways to build the ecological context of the architectural environment and the characteristics of the natural environment, focusing on the physical properties of space, from the perspective of biophilic design. The present study deals with the service of the biophilic experience through connection with a smart-home technology. This study focuses on the immersive experience in order to experience the characteristics of place in a positive natural environment, including various technical solutions that support the physical environment. The experience of space and place is based on the relationship between man and nature and can be made through place-based relationships and evolved human–nature relationships [8]. Place-based relationships are closely related to the ecosystem of the region where the occupants live. From the perspective of biophilia, the architectural system refers not to natural systems or a collection of resources, but a desire for cooperation between man and nature. The collection-to-collaboration approach provides beneficial advantages for both man and nature and enables the conservation of indigenous ecosystems as well as the integration of the local environment [91]. The "Latro lamp", developed by Mike Thompson, fulfills the interrelationship of photosynthesis and energy generation through the extraction of energy during the photosynthesis of algae in the lamp, requiring sunlight, $CO_2$, and water [92]. This is also related to biomimicry that understands and mimics the unique properties of photosynthesis. Phillips Design's "Microbial Home", a service that minimizes waste through periodic input/output systems, systematically creates a circular ecosystem for residential activities, such as supplying energy obtained from waste generated in each residential space to other residential spaces for electricity, etc. [91]. The "methane bio-digester" kitchen island cultivates bacteria by crushing food waste and the excrement from the "filtering squatting toilet" and collecting bacterial gas (biogas) used in built-in lighting and cooking equipment. The "paternoster", a plastic waste decomposition system that utilizes the enzymes of fungus, produces edible mushrooms through plastic decomposition products. These bio-friendly home systems can solve waste treatment and renewable energy issues, reduce the workload and maintenance costs in the elderly's homes, and increase occupants' perception of interdependence with nature by recognizing the ability and circulation system of nature in daily life.

Their evolved relationship with nature indicates a desire in humans to explore and discover the natural environment as well as an interest in all the complex processes of it [23]. Assuming that experiencing the natural environment "as is" can provide direct benefits to the health and well-being of the elderly, there is a need to make this accessible from anywhere, regardless of physical, spatial, or temporal constraints. In particular, human genetic instincts for the natural environment enable

the biophilic experience both sensationally and neurologically through a properly simulated natural environment [93]. The immersive experience, which represents the fusion of sophisticated computer graphics technology, big data transmission technology, and various human-centered interaction technologies [94], can maximize the biological reactions of man to the natural environment and can be used anytime, anywhere through wearable devices, applications, etc. The immersive experience has begun to develop in the fields of games and entertainment, and in recent years, has been widely used in physical exercise, programs that help strengthen cognitive function, and tests of immersion, among others [55,95–97]. In addition, it can also act as a medium to promote the improvement of sensory and motor functions, perceptual ability, cognitive function, formation of bonds, emotional relaxation, etc. [55,85,98,99]. Rendever provides customized VR content for seniors in homes and senior living centers while providing them with various services, such as reminiscing about places, displaying the destinations they wanted to go, and experiencing the natural environment [100]. The elderly, facility officials, and their family members have shown a high level of satisfaction with this as it allows them to experience natural environments through immersive experiences and to interact with family and friends in real time in a virtual environment. As immersive content has been utilized in various fields, such as learning, medical care, and communication, various actuators capable of working with VR/AR devices have been developed [85]. A haptic actuator that supports tactile sensation allows users to more realistically interact with objects they want in virtual and augmented realities; they are becoming thinner and smaller and can be worn as a soft material such as clothing, rather than as mechanical equipment [101]. The immersive content for the biophilic experience needs to be carefully set up so it can accurately represent detailed and diverse natural environments, including famous landmarks, landscapes, major climates, and distinguishing animals and plants [23]. In addition, it is important to express the unique characteristics of specific ecosystems, such as mountains, valleys, forests, wetlands, etc., and an environment with abundant resources and opportunities should be created [19]. For immersive content for the elderly in the future, content for their physical and cognitive training, development of actuators for interaction, etc., can be considered by utilizing the abundant information, discovery, and motivational characteristics of the natural environment [73]. Table 7 lists smart-home services and technologies for the experience of space and place.

**Table 7.** Smart-home services for indirect experience of nature.

| Smart-Home Services | Devices and Sensors | | Literature Source |
|---|---|---|---|
| Nature-immersed contents | VR/AR display | Wearable devices (HMD, EGD, etc.) | [73,85] |
| | | Screen display (FogScreen, wall display, invisible display) | [83,86] |
| | Smart devices and system | PC/mobile devices | [60,62] |
| | | Haptic actuator | [85,101] |
| Collaboration system with nature | Smart devices and system | Smart kitchen and toilet | [82,91] |
| | | Natural energy reproduction system (NERS) [1] | [91,102] |
| | | Rainwater recycling system (RRS) [2] | [79,81] |

[1,2] Most of the paper sources for the elements of smart-home devices were literature review papers. The terms of devices were attuned to maintain consistent terminology, and these were newly established in this study.

## 4. Biophilic Experience Based Smart-Home Service Framework

### 4.1. Smart-Home Service Content for Biophilic Experience

Based on the literature analysis, this study derived the contents of the biophilic experience-based smart-home service, as shown in Table 8.

**Table 8.** Biophilic experience-based smart-home service contents.

| Type of Service for Biophilic Experience | | Service Contents |
|---|---|---|
| Maximization of light | S.1 | Adjustment of louvers and curtains according to the amount of sunlight and its direction |
| | S.2 | Real-time tracking of sunlight paths and inflow of reflected light |
| Pleasant air and thermal | S.3 | Induction of natural ventilation and airflows (wind) according to the wind volume and direction |
| | S.4 | Automatic temperature/humidity control according to the weather |
| | S.5 | Confirmation of indoor air pollution levels and automatic ventilation |
| | S.6 | Greenwall automation management |
| Animals and plants | S.7 | Provision of social robots in the shape of cats or puppies |
| | S.8 | Automatic management of water quality and temperature, food, etc., in water tanks |
| | S.9 | Real-time plant status confirmation and automatic management |
| View and weather | S.10 | Provision of glass that occupants can remotely covert it to transparent one |
| | S.11 | Provision of videos showing real-time information on external weather and conditions |
| | S.12 | Rainwater collection and providing the sound of rain |
| Virtual and augmented nature | S.13 | A virtual display window showing the occupants' desired natural scenery |
| | S.14 | A skylight display showing a virtual sky |
| | S.15 | Provision of simulation of seasonal environments according to climate and of natural sounds |
| | S.16 | Provision of virtual water objects/sounds such as waterfalls, waves, etc. |
| | S.17 | Provision of 3D virtual animal objects (bird, butterfly, dolphin, etc.) |
| | S.18 | Indoor lighting control according to the simulation of virtual objects and background |
| | S.19 | Provision of natural sounds according to the situation of occupants such as sleeping, waking up, eating, etc. |
| | S.20 | Virtual natural wall patterns and textures that can be controlled |
| Simulating natural light and airflow | S.21 | Provision of artificial sunlight according to the intensity of illumination |
| | S.22 | Provision of various virtual light shapes and shadows |
| | S.23 | Provision of HVAC controlled according to occupant status |
| Nature-immersed contents | S.24 | Provision of VR/AR content that users can experience and travel the natural environment |
| | S.25 | Provision of content for physical/cognitive training based on natural environments |
| | S.26 | Virtual natural elements that respond to touches and gestures |
| Collaboration system with nature | S.27 | Waste collection and natural disposal |
| | S.28 | Natural energy reproduction and rainwater recycling |

The present study identified 28 total biophilic-experience-based services, prepared in consideration of the support of direct, indirect, local, or regional experiences between the elderly and nature, based on the technologies and services proposed in previous studies. The maximization of light maximizes the amount and intensity of sunlight entering the room by monitoring the external environment through the heliostat and luminance sensors and through the interaction of the indoor environment controller and actuator according to the collected information. In order to achieve a pleasant air quality and a comfortable thermal environment, namely "Pleasant air and Thermal", indoor natural ventilation is induced through external environment monitoring; the HVAC system automatically adjusts the indoor environment according to the weather to respond to changes in the external environment.

Animals and plants create an environment in which they can communicate with living organisms in houses, supporting the elderly and easily managed. The service "View and weather" protects the privacy of residents, allows them to view the natural environment and improves their perception of the natural environment by providing weather information and rainwater collection information in real time. Virtual and augmented nature and simulating natural light and airflow are technical supports for the indirect experience of the natural environment, providing simulated natural elements using virtual and augmented reality and a more dynamic lighting environment and optimized air quality. Nature-immersion contents allow for self-immersion in a virtual natural environment, and physical and cognitive reinforcement training is possible using the empirical characteristics of nature. Immersive content has excellent interactivity with users and can satisfy both the visual, tactile, and hearing senses, simultaneously, allowing a multi-sensory experience of the relationship between nature and humans. Finally, the collaboration system with nature is based on the cooperation between natural systems and humans and supports the expanded experience of biomimicry. It is possible to recall the interdependent human relationship with nature and to further integrate the local environment and preserve the indigenous ecosystem by directly participating in the natural processing method or through the creation process of resources.

### 4.2. Framework of Biophilic Experience Services

Smart-home services can be configured in various types through a combination of various sensors and devices in the house and Internet of Things (IoT) devices. In order to configure a smart-home network, the following items are required: a home gateway, an IoT cloud, and a resource collaboration-based home platform [103–105]. Figure 4 shows the framework flow for a biophilic experience-based service.

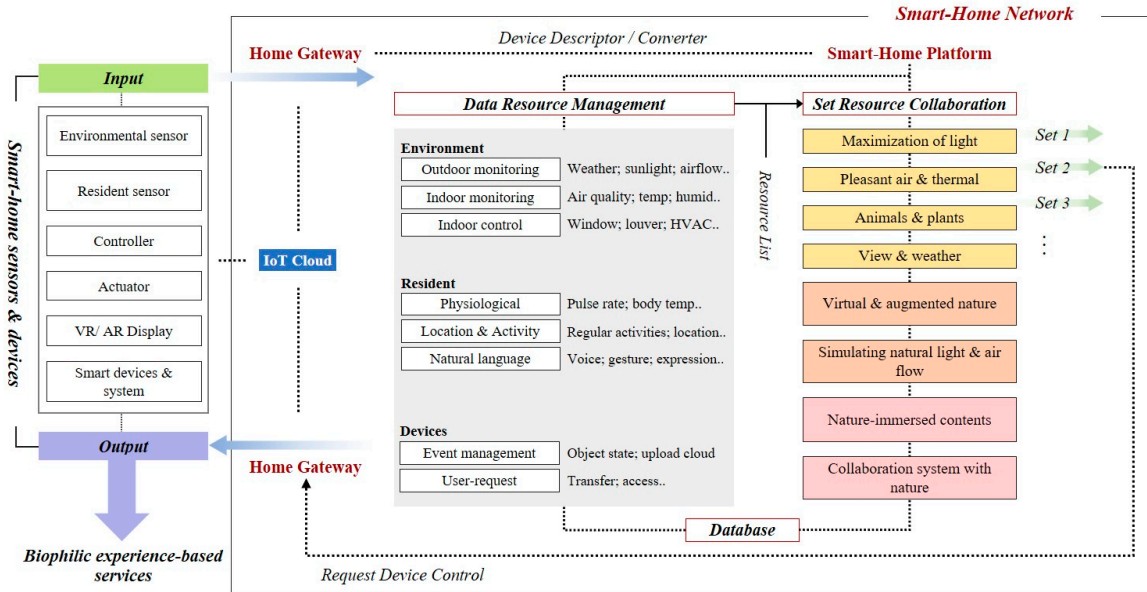

**Figure 4.** Framework flow for biophilic experience-based service.

Our proposed smart-home service framework uses sensors and devices that support the biophilic experience and IoT-based smart devices as service resources, while devices that can monitor and control status information in the house are used through a home gateway, and IoT-based devices are configured to be controlled through the IoT cloud. The smart-home platform manages device information transmitted through the smart gateway and each IoT cloud and converts and integrates protocols of all devices into resources for smooth service provision and collaboration [103,104]. The home gateway and IoT cloud mutually communicate, and all devices delivered to the smart-home

platform through the home gateway are converted to follow the same protocol and classified into environment/resident/device resources according to their functions. Environment and residents are resources for environmental monitoring, environmental control, and resident information recognition inside and outside the house, and the devices are comprised of the resources used for the interaction with the IoT cloud to provide services that are input or output to IoT-based smart devices. The resource list is delivered to set resource collaboration (SRC), thereby making integrated control and additional collaboration possible by users or administrators. That is, once the SRC is completed, the data resource of the input device is defined accordingly, and all repeated processes are stored in the database of the smart-home platform.

This study organized the input and output devices that support the biophilic experience service according to the service framework, as shown in Table 9. Moreover, we proposed the interaction characteristics of smart-home devices from the perspective of residents and space, in consideration of efficient service provision and physical application in the house. From the perspective of residents, the interaction method was analyzed based on whether direct interactions such as physical contact and physical manipulation occur, and the spatial viewpoint is related to the shape of the device and the installation method in the space, so that the user can visually recognize the device. That is, the interaction is classified based on whether the device is attached to the existing environment, invisibly embedded into the space, or integrated in the form of a home appliance to replace the existing one (i.e., M, SE, or EE).

The interaction characteristics of smart-home service devices from the perspectives of residents and spatial considerations are referred to as contact-SE, contact-EE, noncontact-M, noncontact-SE, and noncontact-EE, excluding contact-M. Contact-SE are devices that require direct interaction by residents and are embedded in existing facilities or spatial structures, including smart kitchens and toilets for waste collection and disposal, smart windows with built-in touch screens, and wall displays. Contact-EE are smart devices or autonomous objects that require the physical manipulation of residents, including smart phones that provide remote control applications, natural language recognition, and social robots that can communicate. Noncontact-M are devices in the form of attachments and installations connected to IoT devices or remote control, including environmental monitoring, which performs functions on its own in the existing environment. These include devices such as heliostat sensors and mirrors, VR/AR screen displays, 3D surround-sound speakers, and home systems such as rainwater collection systems. Noncontact-SE devices automatically perform functions when events occur through environmental monitoring, such as indoor control devices and actuators, and are embedded into spatial structures such as windows, glass, and artificial sunlight. Finally, non-contact-EE devices, which are the most prevalent, are built into lighting and appliances for environmental monitoring, such as luminosity sensors, $CO_2$ sensors, and video/microphone sensors, and consist of various sensors and controls, such as a smart aquarium and smart plant growers.

In a prior study on the use of smart-home services by the elderly, the elderly responded positively to devices that are highly preferred for automated smart-home systems and are capable of intuitive interaction and physical manipulation [60,70,106]. Therefore, it is important to provide smart-home services from an integrated perspective. The way in which residents interact with smart-home devices should be adjusted according to the residents' physical and mental characteristics and should support cost savings and space efficiency through intelligent and smart devices that provide a variety of functions in place. The service-assisted devices and interaction characteristics proposed in this study distinguish active and passive use of residents by input and output devices and contribute to the search for physical methods of applying smart-home devices in the residential space.

**Table 9.** Interaction characteristics with input and output devices of service.

| Biophilic Experience Services | | Input Devices | | | Output Devices | | |
|---|---|---|---|---|---|---|---|
| | | Devices | Interaction | | Devices | Interaction | |
| Maximization of light | S.1 | Luminance/Heliostat sensor | Non-contact | EE/M | Controller and actuator | Non-contact | EE |
| | S.2 | Heliostat/mirror | Non-contact | M | Window actuator | Non-contact | SE |
| Pleasant air and thermal | S.3 | Air volume/direction | Non-contact | M | Window actuator | Non-contact | SE |
| | S.4 | Temp/humid sensor | Non-contact | EE | HVAC controller | Non-contact | EE |
| | S.5 | **CO2/TVOC/dust** | Non-contact | EE | HVAC controller | Non-contact | EE |
| | S.6 | GIAS | Non-contact | EE | GIAS | Non-contact | EE |
| Animals and plants | S.7 | AI social robot | Contact | EE | AI social robot | Contact | EE |
| | S.8 | Smart aquarium | Non-contact | EE | Smart aquarium | Non-contact | EE |
| | S.9 | Smart plants growers | Non-contact | EE | Smart plants growers | Non-contact | EE |
| View and weather | S.10 | PC/smart phone | Contact | EE | Smart glass | Non-contact | SE |
| | S.11 | Video/mike outdoor | Non-contact | EE | Screen display | Non-contact | M |
| | S.12 | Video/mike outdoor | Non-contact | EE | RHS/3D surround speaker | Non-contact | M |
| Virtual and augmented nature | S.13 | Smart window | Contact | SE | Smart window | Contact | SE |
| | S.14 | PC/smart phone | Contact | EE | Smart window | Contact | SE |
| | S.15 | Video/mike outdoor | Non-contact | EE | VR·AR Screen/3D surround speaker | Non-contact | M |
| | S.16 | PC/smart phone | Contact | EE | VR·AR Screen/3D surround speaker | Non-contact | M |
| | S.17 | PC/smart phone | Contact | EE | Hologram projector/EGD | Contact | EE |
| | S.18 | VR/AR screen display | Non-contact | M | Smart guide lighting | Non-contact | EE |
| | S.19 | Physiological sensor/LiDAR | Non-contact | EE | 3D surround speaker | Non-contact | M |
| | S.20 | PC/smart phone | Contact | EE | Wall display | Contact | SE |
| Simulating natural light and airflow | S.21 | Luminance sensor | Non-contact | EE | Artificial sunlight | Non-contact | SE |
| | S.22 | PC/smart phone | Contact | EE | Projector lamp | Non-contact | M |
| | S.23 | Physiological sensor/LiDAR | Non-contact | EE | HVAC system | Non-contact | EE |

**Table 9.** *Cont.*

| Biophilic Experience Services | | Input Devices | | | Output Devices | | |
|---|---|---|---|---|---|---|---|
| | | Devices | Interaction | | Devices | Interaction | |
| Nature-immersed contents | S.24 | PC/smart phone | Contact | EE | HMD/EGD, haptic actuator | Contact | EE |
| | S.25 | PC/smart phone | Contact | EE | HMD/EGD, haptic actuator | Contact | EE |
| | S.26 | Multimodal sensor | Contact | EE | HMD/EGD, haptic actuator | Contact | EE |
| Collaboration system with nature | S.27 | Smart Kitchen/toilet | Contact | SE | NERS | Non-contact | M |
| | S.28 | NERS/RRS | Non-contact | M | Lighting, smart plants grower, etc. | Non-contact | EE |

Note: M = mounted type, EE = equipment embedded type, SE = space embedded type.

## 5. Conclusions

This study has suggested a smart-home service framework for the biophilic experience by analyzing previous studies related to smart-home and smart-technology cases that support biophilia. It also provides a theoretical basis for integrating the biophilia concept with the smart-home environment. The conclusions of this study are as follows:

First, when the urban environment is cut off from nature it can degrade human biological capacity, resulting in everyday life becoming more psychologically debilitating. Thus, measures are needed to actively provide the benefits of nature in homes for the healthy aging of the elderly. Previous studies related to biophilia and biophilic design have systematically developed physical planning measures for health, but constraints of local and urban planning tend to overlook day-to-day places in modern society and urban environments [34]. The results of a prior study examining human positive responses in simulated natural environments suggest the potential of smart homes and IT technology for biophilic design [20–22].

Second, the biophilic-experience-based smart-home service proposed in this study was derived from the analysis of the direct, indirect, place, or regional characteristics of the positive biophilic experience for the elderly and the technical factors of the smart home that support it. Smart-home services for the elderly's biophilic experience can be provided through existing smart-home technologies, such as environmental monitoring and control, lighting technology, and home smart devices. However, the purpose of the service must be focused not only on energy efficiency, but also on the exposure to nature according to different situations of residents. In addition to providing energy information for buildings, service resources are needed to deliver weather information and natural soundscapes outside the house. In addition, it is necessary to expand the scope of VR and AR technology and immersive content and use them as a means of natural immersion experience that is free from spatial and temporal constraints, not just a means of providing information. In the case of a home system, it is necessary to develop a cooperative relationship with nature, providing new values of sustainable residential spaces from the perspective of the Green New Deal.

Third, this study proposed a biophilic-experience-based service framework and interaction characteristics for the application of input and output devices and services accordingly. In this study's smart-home service framework, sensors and devices that support a biophilic experience and IoT-based smart devices are used as service resources, and collaboration within resources is possible through a home gateway and an integrated home platform. Specifically, as a result of analyzing the characteristics of the interaction with smart-home devices from the perspective of residents and spatial considerations, it is necessary to consider the reduction of the device implementation cost and space efficiency through resource integration and collaboration according to the interaction method and function.

This study underlines that smart-home technology should reinforce the experience with nature for a sustainable living environment for the elderly; it also identifies methods that link the biophilic experience with smart-home technology. In this process, it is meaningful in that it organized the existing literature and evidence and suggested a smart-home service and utilization plan from a new perspective. Planning techniques that reinforce biophilic tendencies in indoor spaces can be a solution to many health problems of the elderly, but previous studies related to smart homes for the elderly have focused on the treatment and management of diseases, emergency response, and convenience of the elderly rather than providing sufficient support for positive experiences of the elderly in houses or for rapport with nature. In addition, since studies linking the biophilic experience with smart technology and services in the field of biophilic design are insufficient, this study is unique in that it offers a new perspective for enabling the biophilic experience.

This theoretical contribution broadens the understanding of how to experience nature from the perspective of a sustainable residential environment. Moreover, it has value in managing positive experiences and maintaining a high QoL in the elderly in smart homes. In particular, the biophilic experience-based smart-home services content proposed in this study informs the expansion of the aged-friendly smart-home industry and contributes to the development of smart-home services along

with new service trends. Finally, this study contributes to the consideration of improving the physical applicability of devices in smart homes and the composition of resource collaboration of devices for each service by proposing the interaction characteristics of input and output devices according to the service framework, residents, and spatial perspectives. As the biophilic experience service and framework of the study utilizes the limited case category of previous studies included in the literature review, one limitation of this work is the presentation of a clear criteria between certain items in the range of analysis results and the suggestion of interaction characteristics. This is because there are cases in which the definition of a single analysis criterion is vague because of the diversification and complexity of the contents of services; however, this study is meaningful in that the service plan has taken a holistic approach to services and devices, occupants, and spaces. In particular, device type and interaction method are related to the physical factors and usability of the installation method, as they can be used in related research fields. In further studies, the quantitative data will be analyzed using a clear and rigorous methodology, and the search criteria will be expanded. In particular, for the implementation of the framework suggested in Figure 4, it is important to consider technical alternatives and specific implementation methods, and to proceed with convergence studies that take them into account. Overall, it is necessary to provide further suggestions for ways to alleviate the restrictions and limitations on the use of smart-home services for the elderly. It is also crucial to gauge the importance of the elderly's biophilic experience services in the house, considering the elderly's satisfaction and physical and cognitive functions.

**Author Contributions:** E.J.L. and S.J.P. conceived and designed the research; E.J.L. collected and analyzed information by reviewing the literature and investigating case; S.J.P. consistently examined, modified, and supplemented the manuscript. All authors have read and agreed to the published version of the manuscript.

**Funding:** This research was funded by NATIONAL RESEARCH FOUNDATION OF KOREA (NRF) under the Korean Government Ministry of Education, Science and Technology (MEST), grant number 2018R1C1B6008735; This research was funded by Basic Science Research Program through the NATIONAL RESEARCH FOUNDATION OF KOREA (NRF) funded by the Ministry of Education, grant number 2020R1A6A3A13077228.

**Conflicts of Interest:** The authors declare no conflict of interest.

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
