# Peer review of "A Framework of Smart-Home Service for Elderly’s Biophilic Experience"

_sustainability, doi:10.3390/su12208572_

Round 1

Reviewer 1 Report

The manuscript entitled “A Framework of Smart-Home Service for Elderly’s Biophilia Experience” presents a theoretical basis about biophilia and smart-home technologies and an associated framework to the biophilia experience in smart-home technology and services. It is worth highlighting the emergence and importance of this topic and the way authors organized the existing evidence and presented a series of practical strategies.

I would like to notice that the topic and aim are well defined and that the results clearly provide an advance in current knowledge. Despite these very positive aspects, I consider that the scientific soundness can be improved since the procedures of review are not clearly presented, namely the number of results from the search, number of selected and rejected papers. Also about the quality of presentation, the authors wrote and delved so deeply that the article is difficult to analyze and follow.

Regarding more specific aspects to correct, here is a list of some to consider:

  • Page 1, line 40 – the indication of the reference (8) should only appear at the end of the sentence.
  • Page 7, line 196 - When presenting data from the reference 1, authors only refer Wilson but the first author of this paper is Kellert.
  • Page 10, line 327 – Authors refer “Chapter 2”, which is not appropriate to article´s structure. “Chapter” is more associated to books.
  • Page 10, line 328 – Information about the reference of Kellert is missing.
  • Page 10, figure 2 – Check if is “amenity” or “awareness” improvement.

Author Response

Reviewer 1

Point 1: The manuscript entitled “A Framework of Smart-Home Service for Elderly’s Biophilia Experience” presents a theoretical basis about biophilia and smart-home technologies and an associated framework to the biophilia experience in smart-home technology and services. It is worth highlighting the emergence and importance of this topic and the way authors organized the existing evidence and presented a series of practical strategies.

Response 1: Thank you very much for reviewing the paper and providing useful comments- your opinion is highly valued. Conclusions have been supplemented to highlight the importance and significance of this research (page 21, line 632-634).

Point 2: I would like to notice that the topic and aim are well defined and that the results clearly provide an advance in current knowledge. Despite these very positive aspects, I consider that the scientific soundness can be improved since the procedures of review are not clearly presented, namely the number of results from the search, number of selected and rejected papers. Also about the quality of presentation, the authors wrote and delved so deeply that the article is difficult to analyze and follow.

Response 2: We have added Figure 3 in section 3.3 and added the relevant contents in the manuscript in accordance with the comments.

Point 3: Regarding more specific aspects to correct, here is a list of some to consider:

Page 1, line 40 – the indication of the reference (8) should only appear at the end of the sentence.

Response 3: This has been modified.

Point 4: Page 7, line 196 - When presenting data from the reference 1, authors only refer Wilson but the first author of this paper is Kellert.

Response 4: We have reconstructed and supplemented the sentence in context.

Point 5: Page 10, line 327 – Authors refer “Chapter 2”, which is not appropriate to article´s structure. “Chapter” is more associated to books.

Response 5: We have modified “Chapter” to “Section” to reflect the detailed comments.

Point 6: Page 10, line 328 – Information about the reference of Kellert is missing.

Response 6: This has been modified.

Point 7: Page 10, figure 2 – Check if is “amenity” or “awareness” improvement.

Response 7: As recommended, we considered using the term “awareness.” However, that services are intended to support the daily life and convenience of the elderly in the home. Thus, we came to the conclusion that “amenity” was more appropriate.

Reviewer 2 Report

Dear editor,
The manuscript has great originality due to the theme provided. High degree of detail in all its sections and  the great relevance and demand of the subject. In addition to the number of downloads of it due to being current. The work presents the benefits of the application of technologies in physical and mental health, but not only that; but how to interact  through virtual from residences with the environment, and its health benefits.
Regards

Author Response

Point 1: Dear editor,
The manuscript has great originality due to the theme provided. High degree of detail in all its sections and the great relevance and demand of the subject. In addition to the number of downloads of it due to being current. The work presents the benefits of the application of technologies in physical and mental health, but not only that; but how to interact through virtual from residences with the environment, and its health benefits.
Regards

Response 1: We appreciate your opinion, and are grateful for your positive review of this paper.

Reviewer 3 Report

In this study, the authors suggest a framework for smart-home services able to support the biophilia experience. After the literature review, the paper provides theoretical insights on integrating biophilia concepts with the smart-home environment.

Overall the study is interesting and informative; it will be of interest to the readership without any doubt.

A minor suggestion is that it would be useful to see some discussion about technical aspects (e.g., software engineering, sensor/signal processing, data mining, etc.) for implementing the framework presented in Figure 3.

The English can be improved by a native-English-speaking person.

Author Response

Point 1: In this study, the authors suggest a framework for smart-home services able to support the biophilia experience. After the literature review, the paper provides theoretical insights on integrating biophilia concepts with the smart-home environment.

Overall the study is interesting and informative; it will be of interest to the readership without any doubt.

Response 1: We are grateful for the careful review of this paper, and your opinion is highly valued.

Point 2: A minor suggestion is that it would be useful to see some discussion about technical aspects (e.g., software engineering, sensor/signal processing, data mining, etc.) for implementing the framework presented in Figure 3.

Response 2: This study underlines the benefits and importance of applying technology to housing from the perspective of biophilia and focuses on the applicability of smart home services and related technologies. Thus, considering that the implementation plan of the specific technology proposed in this study is insufficient, the limitations of the study and the direction of future research are included in the conclusion section (page 22, line 658-661).

Point 3: The English can be improved by a native-English-speaking person.

Response 3: In response to this opinion, we have availed the services of a professional proofreading company to improve the overall readability of the manuscript (please refer to the attached file of the editing certificate).

Round 2

Reviewer 1 Report

In relation to the review process of the manuscript "A Framework of Smart-Home Service for Elderly’s Biophilic Experience, I can assert that the authors carefully followed the given recommendations. I do not consider the need for further modifications.